# Integrated Satellite, Unmanned Aerial Vehicle (UAV) and Ground Inversion of the SPAD of Winter Wheat in the Reviving Stage

**DOI:** 10.3390/s19071485

**Published:** 2019-03-27

**Authors:** Suming Zhang, Gengxing Zhao, Kun Lang, Baowei Su, Xiaona Chen, Xue Xi, Huabin Zhang

**Affiliations:** 1National Engineering Laboratory for Efficient Utilization of Soil and Fertilizer Resources, College of Resources and Environment, Shandong Agricultural University, Taian 271018, China; zhangsuming1994@163.com (S.Z.); langkun1201@foxmail.com (K.L.); 13345277378@163.com (B.S.); 18763899725@163.com (X.C.); 18562316568@163.com (X.X.); 2Shandong Huibangbohai Agricultural Development Co., Ltd., Dongying 257091, China; huabinzhang@126.com

**Keywords:** remote sensing, UAV, Sentinel-2A satellite, SPAD, chlorophyll

## Abstract

Chlorophyll is the most important component of crop photosynthesis, and the reviving stage is an important period during the rapid growth of winter wheat. Therefore, rapid and precise monitoring of chlorophyll content in winter wheat during the reviving stage is of great significance. The satellite-UAV-ground integrated inversion method is an innovative solution. In this study, the core region of the Yellow River Delta (YRD) is used as a study area. Ground measurements data, UAV multispectral and Sentinel-2A multispectral imagery are used as data sources. First, representative plots in the Hekou District were selected as the core test area, and 140 ground sampling points were selected. Based on the measured SPAD values and UAV multispectral images, UAV-based SPAD inversion models were constructed, and the most accurate model was selected. Second, by comparing satellite and UAV imagery, a reflectance correction for satellite imagery was performed. Finally, based on the UAV-based inversion model and satellite imagery after reflectance correction, the inversion results for SPAD values in multi-scale were obtained. The results showed that green, red, red-edge and near-infrared bands were significantly correlated with SPAD values. The modeling precisions of the best inversion model are R^2^ = 0.926, Root Mean Squared Error (RMSE) = 0.63 and Mean Absolute Error (MAE) = 0.92, and the verification precisions are R^2^ = 0.934, RMSE = 0.78 and MAE = 0.87. The Sentinel-2A imagery after the reflectance correction has a pronounced inversion effect; the SPAD values in the study area were concentrated between 40 and 60, showing an increasing trend from the eastern coast to the southwest and west, with obvious spatial differences. This study synthesizes the advantages of satellite, UAV and ground methods, and the proposed satellite-UAV-ground integrated inversion method has important implications for real-time, rapid and precision SPAD values collected on multiple scales.

## 1. Introduction

Chlorophyll is an important pigment in plant photosynthesis, as its content reveals the ability of a plant to conduct material energy exchange with the external environment, as well as growth status, primary productivity, carbon sequestration ability and nitrogen utilization efficiency, and so on. Chlorophyll is not only the important index for evaluating crop condition, but it is also the indicator of growth and development stage [1,2]. The reviving stage of wheat is the second tillering peak of wheat, at which point the number of tillers increases by 30–40%. During this period, the chlorophyll content has an important influence on the future growth rate of wheat.

The spectral characteristics of chlorophyll are the most important basis for its content detection. The green and red bands have been shown as the most valuable for detecting chlorophyll [3,4]; however, some studies have also identified the near-infrared channel as an effective candidate [3,5]. Previous studies also show that reflectance spectra in the visible region (400~700 nm) can effectively estimate chlorophyll [6]. At the same time, due to the strong absorption of red light by chlorophyll and the strong reflection of near-infrared light, a steep red-edge is usually formed between 680 and 760 nm. Therefore, the spectral response of the red-edge to chlorophyll is also very strong. The red-edge parameters became one of the most important index for the growth of crops [7], as well as for estimating chlorophyll content in plants [8]. And then, the optimal red-edge parameters were screened out by detecting the hyperspectral values and chlorophyll content, and a relationship model was established for them [9].

There are many methods for measuring chlorophyll content. The traditional methods mainly include ultraviolet and visible spectrophotometry and fluorescence analysis, all of which are used to conduct chemical quantitative analysis of chlorophyll content by means of the substance’s spectral characteristics [10,11]. The results of these methods are accurate but time-consuming and laborious and are destructive to crop leaves. The Soil and Plant Analysis Development (SPAD) method was used for nondestructive and rapid measurements of SPAD values of leaves by means of light and electricity [12,13]; the relative chlorophyll content of leaves is expressed through SPAD values, and the results are almost the same as the results of chemical experiments, which can replace the traditional chemical measurement method. However, this method has limited measuring points and is not suitable for large area measurements [14,15]. 

Satellite remote sensing is a mainstream means of agricultural remote sensing; the red-edge bands of Sentinel series satellites were used to monitor the chlorophyll content of crops [16]; Sentinel-2 imagery were used to monitor the canopy chlorophyll content and obtained a high accuracy [17]. Because satellite remote sensing can provide large-scale, frequent, low cost and massive contents of information, it has gradually replaced traditional methods of SPAD values monitoring, which are inefficient and expensive. However, due to factors such as the fixed orbit of satellites, the long revisit period, the effects of the atmosphere, the low spatial resolution, and so on [18], meeting the needs of high-precision real-time quantitative inversion in the field is difficult [19,20].

In recent years, unmanned aerial vehicle (UAV) technology has gradually entered the civil field and has become a hot spot for practical research and applications in agriculture. UAV remote sensing provides a nondestructive and cost-effective way for rapid monitoring SPAD values because it is easier to build a platform, is able to fly at a lower altitude and has flexible movement areas, and can capture both spatial and temporal high resolution images [21]. UAV technology has been widely used in fine agriculture [22,23,24]. However, there are still some limitations of current UAV technology; for example, coverage of UAV images is limited, so it is not suitable for large-scale applications. At the same time, there is also room for further improvement in inversion accuracy [25].

Scholars have also conducted research on the application of multisource data and generated robust results [26,27]. However, most of them combined a variety of satellite data with ground measured data [28,29,30] or satellite images combined with surface hyperspectral data [31,32]. Accordingly, the integration of satellite, UAV and ground data is rare, feasible and is thus worthy of further research.

In this study, the core region of the Yellow River Delta (YRD) was selected as the study area, the high-precision ground data are measured by a SPAD instrument, and large-scale images of satellite and high spatial-temporal resolution data of UAV were combined to comprehensively solve the problems of real-time, multi-scale and high-accuracy monitoring of winter wheat farmland in the study area. Compared with the traditional measurement method of SPAD values, this study provides a multi-scale, high-efficiency and nondestructive estimation method of SPAD values [33,34], and this method has a higher accuracy and is less affected by the weather compared with being based on the measured data and satellite images [35,36,37]; moreover, when compared with using measured data and UAV images, the estimation scale of this method is larger [38,39]. In this study, the following issues were explored: (1) constructing inversion models of SPAD values based on the relationship between the measured data of the SPAD instrument and the UAV images and electing the best model; (2) calculating the corrections coefficient of satellite imagery based on the consistent relationship between UAV images’ reflectance and satellite images’ reflectance to correct the reflectance of satellite images; (3) applying the best inversion model to the satellite imagery that has been corrected to realize a scale-up inversion of winter wheat in the reviving stage.

## 2. Data and Methods

The data sources in this paper include satellite, UAV and ground measured data. When building the UAV-based model, a ground sampling point data corresponds to a pixel of UAV imagery. When building the satellite-based model, a ground sampling point data corresponds to a pixel of satellite imagery. When calculating the correction coefficient of reflectivity of the satellite images, there are approximately 1600 UAV image pixels corresponding to a satellite image pixel in the same area.

### 2.1. Study Area

The study area is located in the core region of the Yellow River Delta--Hekou District (37°45′~38°10′ N, 118°07′~119°05′ E) and Kenli District (37°24′~38°10′ N, 118°15′~119°19′ E). The study area is in the middle latitudes and is located at the junction of land and sea. The climate is the warm temperate monsoon continental type, where the average annual temperature is approximately 13 °C and the frost-free period is more than 200 days. The climatic conditions meet the requirements of two-year and three-cropping of crops. Cultivated land is the largest land use type in the study area, accounting for 42.8% of the total [40]. The main grain crops are wheat and maize. However, because the study area is located in the coastal plain, the saline land area is large and is widely distributed, which has a negative impact on crop production [41]. Therefore, real-time, multi-scale and high-accuracy monitoring of winter wheat growth is of great importance to agricultural production and economic development in this area.

Bohai farm in the Hekou District was selected as the core test area for the UAV flight test and ground survey (Figure 1). The difference between SPAD values in this core test area is obvious, which lays a foundation for the universality of the inversion model of SPAD values. 

### 2.2. Data Acquisition and Preprocessing

#### 2.2.1. Data Acquisition of SPAD Values

A total of 140 observation sample areas were uniformly distributed in the core test area, and they were selected based on the following factors: (1) the observation sample areas were spatially representative with an identical land use type; (2) growth of crops and the vegetation coverage were uniform; (3) an area of 30 m × 30 m was covered, and SPAD values of wheat leaves were measured by a SPAD 502 chlorophyll meter [42]. Wheat plants were randomly selected from each sample area. The upper, middle and lower parts of five representative leaves were measured twice, and a total of 30 SPAD values were obtained, with the average value being taken as the SPAD value of an observed sample area. Meanwhile, the coordinates (longitude and latitude) of sampling points were measured by a Trimble GEO 7X centimeter hand-held GPS. The sampling dates were 5~10 March 2018.

#### 2.2.2. UAV Imagery Acquisition and Processing

The Sequoia multispectral camera is mounted on a Dajiang Matrice 600 Pro UAV (loaded weight 5.5 kg and flying duration 16 min), which contains four bands: green, red, red edge and near infrared [43,44] (Table 1). On 8 March 2018 at 12:00 to 14:00, the multispectral imagery was acquired by the UAV in the core test area. The weather was sunny, with a temperature range from 4~6 °C and northwest winds of 3~5 m/s, and the angle of the solar incidence was 52°. There were three missions; the flight height was 100 m, the flight speed was 5 m/s, the camera shooting time interval was set to 1.5 s, the area was approximately 1 km^2^, and the flight plan was designed to ensure 80% overlap with the resolution of 4~5 cm. More than 5000 valid images were taken. In the later stage, Pix4D software was used to preprocess the UAV imagery, with such activities as mosaic and radiation corrections (Figure 1D).

#### 2.2.3. Acquisition and Processing of Sentinel-2A 

Sentinel is a series of satellites launched by ESA Copernicus [45]. Among them, Sentinel-2 is a series of optical satellite that contains two satellites--Sentinel-2A and Sentinel-2B [46]. In this paper, Sentinel-2A (S2A) products are downloaded through the ESA Copernicus data sharing website (https://scihub.copernicus.eu/). Images from March 1, 2018 were selected to cover the study area at the same time of shooting for each image.

The downloaded data are LIC atmospheric reflectance data that have been geometrically corrected. By using SNAP software provided by ESA, atmospheric correction, resampling and export of ENVI format data were completed. Then, mosaic, reflectivity extraction, image clipping, and so on were processed in the ENVI 5.1 software.

S2A includes 13 bands. In considering the consistency of S2A with UAV and the requirements for model building, the satellite bands that are consistent with the range of the Sequoia were selected, as shown in Table 1.

### 2.3. Inversion Model of SPAD Values Based on UAV Imagery

The SPAD values and the UAV imagery pixels were individually corresponded and divided into a modeling set and validation set at a ratio of 2:1. First, a correlation analysis between all ground and UAV data was used to screen out the characteristic bands of SPAD values. At the same time, the relationship between SPAD values and various vegetation indexes was analyzed, and the best vegetation indexes that reflected SPAD in the study area were selected as sensitive vegetation indexes and used in the construction of an inversion model. Correlation analysis can measure the degree of correlation between two variable factors, the index is the correlation coefficient (expressed as R), which is a measure of the degree of linear correlation between variables. The larger the absolute value of R is, the higher the correlation between variables will be [47]. In this paper, the Pearson correlation coefficient is adopted as it is the most commonly used method. Vegetation indexes are shown in Table 2. 

Then, new spectral parameters were generated, including multiband information by bands or vegetation indexes combinations (adding, subtracting, multiplying and dividing operations between bands) which are highly correlated with SPAD values and are chosen as the characteristic spectral parameters. Finally, the characteristic bands, sensitive vegetation indexes and characteristic spectral parameters of the modeling set were used as dependent variables to construct the SPAD inversion model by using various regression analyses. Moreover, the validation data is used for the model validation, which brought the spectral data of each sampling point in the verification set into the models to obtain the SPAD inversion values for each sampling point; this process compared the inversion values with the measured values and the verification precision was obtained.

The modeling precision of the model was synthesized with the verification precision (represented by decision coefficient R^2^, root mean squared error RMSE and mean absolute error MAE) to select the best inverse model. R^2^ is used to compare and evaluate the performances of the models. If R^2^ is greater than 0.91, the prediction model is considered to be accurate, and an R^2^ between 0.82 and 0.90 indicates a good prediction; however, the R^2^ of an approximate prediction is considered to lie between 0.66 and 0.81, and an R^2^ between 0.50 and 0.65 indicates a poor relationship [48]. RMES represents the degree of dispersion between the data and the true values; the lower the RMES is, the lower the degree of dispersion of the data is, and the higher the accuracy is [49]. MAE can accurately reflect the actual error by avoiding offset errors, the lower the MAE is, the lower the error of the model is, and the higher the precision is [50,51]. Relevance analysis and model building are performed by software SPSS 22 (Statistical Product and Service Solutions) and WAKE (Waikato Environment for Knowledge Analysis).

### 2.4. Reflectance Correction of S2A imagery

The average reflectivity of each band of the 1600 pixels of UAV images and the corresponding pixel of S2A images is obtained, and the relationship between them is analyzed. The correction method called “mean of ratio” was used to correct the reflectance of S2A [52]. First, the ratios of reflectance for each band of the UAV and S2A (such as B3−Green/Green) were calculated. Then, the means of all ratios as the reflectivity correction coefficients of each band of S2A imagery were taken.

To obtain the corrected S2A imagery, the reflectance of each band of S2A imagery was divided by the reflectivity correction coefficients, and the process is implemented in ENVI 5.1.

### 2.5. Verification of SPAD Inversion Model

Whether the SPAD inversion model based on the UAV image data and ground SPAD values can be applied to S2A images following the corrected reflectance, and how effective the inversion is, must be verified and analyzed. For this reason, three different cases were compared and analyzed for accuracy: applying UAV-based inversion model to S2A imagery before and after the reflectance correction and the S2A-based model applied to S2A imagery. Thus, the effectiveness and feasibility of the satellite imagery reflectance correction method is verified.

### 2.6. Extraction of Winter Wheat Planting Areas

In the spectral analysis of winter wheat in the reviving stage in the study area, the wheat responded strongly to red (Br) and near-infrared (Bnir) bands, while other land features in the study area responded weakly to Br and Bnir bands. Therefore, the NDVI vegetation index threshold method [53] was used in this paper to extract the winter wheat planting area from the S2A image on March 1, 2018. According to the spectral characteristics of the image and the field investigation results, the threshold was determined to be 0.16~0.5, i.e., the mask operation was performed for regions with 0.16 < NDVI < 0.5 (most of the areas with an NDVI > 0.5 were ornamental evergreen tree forests in urban areas), and an image of the winter wheat planting area on March 1, 2018 was obtained.

### 2.7. Inversion of SPAD Values

#### 2.7.1. Inversion of SPAD in the Core Test Area

Based on UAV images obtained on 5~10 March 2018 and the best inversion model, an inversion map of the SPAD values was obtained, and the process is implemented in ArcGIS 10.1 and ENVI 5.1. Those results were compared with the actual situation and measurements in the field.

#### 2.7.2. Scale-up Inversion of SPAD Values in the Study Area

S2A imagery in the study area on 1 March 2018, after image preprocessing and reflectivity correction were used to conduct a SPAD upscale inversion of the study area as based on the optimal SPAD inversion model. The scale-up inversion of SPAD values in the core region of the YRD was carried out, which realized a real-time, rapid monitoring of SPAD at a large scale. The process is implemented in ArcGIS 10.1 and ENVI 5.1.

## 3. Results and Analysis

According to these methods and using the previously mentioned materials, the results obtained in this paper are as follows.

### 3.1. Screening of Characteristic Bands, Sensitive Vegetation Indexes and Characteristic Spectral Parameters of SPAD

#### 3.1.1. Characteristic Bands

The correlation between SPAD values and the reflectance of UAV images and satellite images is shown in Table 3. The reflectance values of the Bg, Bre Breg and Bnir bands of UAV images are positively correlated with the SPAD values, and the Br band was negatively correlated with SPAD values. All bands were significantly correlated with SPAD values at a level of 0.01. The overall trend of correlation between the reflectivity of each band of satellite imagery and SPAD values is similar to that of UAV imagery; however, the R is low, hence, the inversion model based on UAV imagery will be more accurate than the satellite imagery model. Therefore, based on the UAV imagery, this paper selected the four bands of Bg, Br, Bre and Bnir as the SPAD characteristic bands.

To follow the principle of maintaining consistency with the bands of the UAV imagery, the B3-Green, B4-Red, B6-Vegetation Red Edge and B7-Vegetation Red Edge bands of S2A correspond to the Bg, Br, Bre and Bnir bands of the model, and this schema was applied to the scale-up inversion of satellite images. Notably, the B5-Vegetation Red Edge bands of S2A imagery (represented by ${\mathrm{Bre}}_{-705}$ in the model) have a high correlation with the SPAD values of 0.628. Although this band does not participate in the scale-up inversion, it is applied to the model built based on satellite images.

#### 3.1.2. Sensitive Vegetation Indexes

As shown in Table 4, the correlation between vegetation indexes and SPAD values based on UAV data is much higher than for single characteristic bands. Therefore, NDVI, DVI, SAVI_L=0.5_, OSAVI and TCARI with R greater than 0.9 were selected as the SPAD Sensitive vegetation indexes and were used in the model.

#### 3.1.3. Characteristic Spectral Parameters

Table 5 shows the correlation between UAV-based characteristic spectral parameters and SPAD values. Compared with single characteristic bands, the spectral parameters constructed based on characteristic bands are significantly more strongly correlated, and the spectral parameters based on vegetation indexes were also more strongly correlated than vegetation indexes. In general, the correlation of spectral parameters based on vegetation indexes is higher than for characteristic bands. 

### 3.2. Inversion Model of SPAD Based on UAV Imagery

Based on the UAV imagery, the SPAD inversion model was constructed by different regression method with the independent variables of characteristic bands, sensitive vegetation indexes and characteristic spectral parameters. 

The results shown in Table 6 are inversion models of SPAD based on non-linear regression methods, as follows, support vector regression (SVR), back propagation neural network (BP-NN), partial least squares regression (PLS). And the results shown in Table 7 are inversion models of SPAD based on one-variable linear regression (OLR) and multiple linear regression (MLR) methods. Objectively speaking, models based on non-linear methods have good prediction ability, lower error and stronger stability. SVR has the highest accuracy.

As for the linear regression methods, the modeling precisions of MLR, are higher than that of OLR. The modeling R^2^ of the models using MLR, based on the three types of variables, are above 0.8, which indicate high prediction ability; however, MAE and RMSE show significant differences due to different independent variables. Overall, the model constructed with spectral parameters as independent variables, based on MLR, has the highest modeling R^2^, the lowest modeling RMSE and MAE, which shows high accuracy and stability. The same trend is also reflected of verification precisions. Therefore, the model constructed by using MLR method with spectral parameters as independent variables is the best.

Comparing the inversion models based on SVR and MLR method, although the modeling R^2^ of the best MLR model is worse than that of SVR model, its RMSE and MAE indicate a lower prediction error. And the verification precisions of the best MLR model are better than SVR, which show a stronger stability. Meanwhile, for practical application, the MLR model is simpler, more visible and can provide better experimental repeatability, while encapsulated SVR does not have these advantages. Therefore, no matter from accuracy of predictive, practicability, and experimental repeatability, MLR method is the best choice. The best SPAD inversion model is Y=46.803+79.564×G×R−35.583×G×R×REG+33.52×(DVI+TCARI)−9.308×(OSAVI+TCARI) (where Y represents the inversion values of the SPAD of winter wheat in reviving stage) with the modeling precisions of R^2^ = 0.926, RMSE = 0.63 and MAE = 0.92, and the verification precisions of R^2^ = 0.934, RMSE = 0.78 and MAE = 0.87.

### 3.3. Reflectance Correction of S2A imagery

#### 3.3.1. Comparison of Surface Reflectance Between UAV and S2A Imagery

Figure 2 compares the surface reflectance of UAV imagery with S2A imagery. Figure 2a shows the surface reflectance of UAV imagery is higher than the S2A imagery, but the change trend of both images is basically the same; Figure 2b shows a strong correlation between them.

#### 3.3.2. Reflectance Correction Coefficients of S2A Imagery 

The reflectance corrected coefficients of each band are obtained from the ratio of reflectance of UAV and S2A imagery (Table 8). The reflectance of each band of S2A images are divided by the correction coefficients of the corresponding bands, and the corrected S2A images are obtained.

### 3.4. Validation of SPAD Inversion Model

Table 9 compares the accuracy in three cases: (1) the inversion model of SPAD based on S2A imagery is applied directly to S2A imagery; (2) the best SPAD inversion model based on UAV imagery is applied to S2A images before the reflectance correction and (3) after the reflectance correction.

Table 9 shows that the modeling precision based on S2A data (case 1) is lower than for UAV data (case 2 and 3); case 2 has the lowest validation precision, which indicates that the model based on UAV data cannot be directly applied to the S2A images without a reflectivity correction; case 3 has the highest modeling precision and validation precision, which proves that the UAV model itself is of high stability and universality. In summary, applying the best inversion model of SPAD based on UAV imagery to S2A images after the reflectance correction has the best effect and is more feasible.

### 3.5. Extraction Results and Precision of Winter Wheat Area in the Study Area

Table 10 shows the comparison between the extracted area of winter wheat planting and statistical area and the precision of extraction. Because the statistical yearbook of 2018 was not published at the time of investigation, the average planting area of winter wheat in 2016 and 2015 in the statistical yearbook of 2017 was used as the reference values to verify the extraction area of the winter wheat area in 2018.

The results show a high precision and that the NDVI threshold method has high accuracy and feasibility for the area extraction of winter wheat in the reviving stage in the study area. Figure 3 is the distribution map of winter wheat planting area extracted based on S2A images on 1 March 2018 and the vegetation index threshold method. By comparison with the land use status map of the study area, the spatial distribution of wheat is in line with the reality.

### 3.6. Inversion of SPAD values

#### 3.6.1. Inversion of SPAD Values in the Core Test Area

Figure 4 is the inversion of SPAD values in a part of the core test area (0.1 km × 0.5 km). The spatial distribution map of SPAD values in the core test area is obtained after classification. Table 11 is the proportion of areas with SPAD values in different ranges.

The SPAD values for the wheat canopy in the core test area are mostly concentrated in the range of 40~50 and are distributed throughout the whole area, accounting for 61.913% of this map. The second most range is 50~60, which is mainly distributed in the middle-east and southwest of the core test area, accounting for 30.823%. Most of the wheat with a SPAD > 60 was concentrated in the southwest, southeast and northeast parts of the field, while the wheat with SPAD < 40 was less widely distributed, being mainly distributed in the northeast part of the core test area. In general, the inversion results are consistent with the actual situation, proving that the model has high inversion accuracy at the field scale and can accurately invert the SPAD values in the core test area.

#### 3.6.2. Inversion of SPAD Values in the Study Area

Figure 5 is the inversion results of the study area, based on the S2A imagery, and Table 12 shows the classification statistics of SPAD values of winter wheat in the reviving stage in the study area. 

The SPAD values for winter wheat in the reviving stage in the study area were concentrated in the range of 40~50 and 50~60, accounting for 48.98% and 42.29% of the wheat area in the study area, respectively. Overall, the SPAD values showed a gradually increasing trend from the east coast to the northwest and the west. Winter wheat with the highest SPAD values was mainly distributed in the southwest part of Kenli District and the west part of Hekou District, where the overall terrain was higher and the soil salinization was lighter. The areas with lower SPAD values were mainly distributed along the east coast, where soil salinization was serious. The inversion results reflect the spatial characteristics and rules of SPAD values of winter wheat in the study area and have a positive reference value for wheat production management.

## 4. Discussion

(1) Time phase selection is an important part of regional crop monitoring. In this study, the reviving stage of winter wheat in early spring was selected. On the one hand, the reviving stage is the key period for the rapid growth of the winter wheat, and the real-time, rapid and accurate monitoring of its chlorophyll content is of guiding importance for the later production management of winter wheat. On the other hand, this period shows the largest difference in the spectral characteristics of winter wheat and other features in the study area. Monitoring of SPAD values during reviving stage can reduce the noise of extraction and the inversion of SPAD values in winter wheat planting area to the greatest extent and can thus effectively improve research accuracy.

(2) The research shows that the correlation between UAV imagery and SPAD values is generally higher than S2A imagery, which indicates that a high spatial resolution has a significant effect on the recognition of reviving winter wheat spectral information. The centimeter resolution of the Sequoia multispectral camera can effectively remove the influence of mixed pixels and improve the purity of wheat spectral information so that the SPAD inversion model based on UAV imagery has strong universality and high stability.

With the development of the miniaturization of hyperspectral cameras, acquiring hyperspectral images via UAV is now possible [54,55]. However, the sensitive bands of chlorophyll in crops are mainly concentrated in the visible to near-infrared range [56,57], and this study also proves that green, red, red-edge and near-infrared bands are sufficient for realizing the SPAD inversion of winter wheat in the reviving stage in the study area. Therefore, in this experiment, a Sequoia multispectral camera was selected to save cost and reduce data redundancy.

(3) Among the vegetation indexes, TCARI had the highest correlation with SPAD values, followed by DVI and OSAVI. The red and near-infrared bands contribute significantly to the SPAD inversion of winter wheat during the reviving stage, and the addition of the red-edge can further increase the response of vegetation indexes to SPAD values [58]. At the same time, the correlation between OSAVI, SAVI_L=0.5_ and SPAD values is high, which indicates that the vegetation coverage of winter wheat is limited during the reviving stage, and excluding the influence of soil background factors as far as possible in extracting the crop spectral information is necessary. An OSAVI with a parameter of L = 0.16 is more suitable for the removal of soil background effects in this study [59].

(4) In the process of constructing the SPAD inversion model based on S2A data, the participation of the B5-Vegetation Red Edge (${\mathrm{Breg}}_{-705}$) band makes the S2A-based model accuracy reach 0.8, which is acceptable for a general application at a large scale, and it shows that the ${\mathrm{Breg}}_{-705}$ band has a strong correlation with SPAD [60]. Meanwhile, the UAV-based model, as a transition between ground and S2A data, improves the inversion accuracy to greater level, which meets higher accuracy requirements and applications. Therefore, without the support of UAV, the SPAD inversion model based on S2A data constructed in this paper can be directly applied to S2A images after preprocessing for the inversion of SPAD values of winter wheat in the reviving stage. However, different crops and environments have notably different responses to different bands in different growth periods [61,62].

(5) Based on the inversion results of winter wheat SPAD values in the core test area and the study area, the SPAD values of winter wheat have a strong correlation with soil salinity [63]. In the eastern coastal areas with high soil salinity, the SPAD values of winter wheat are generally lower, while in the western and southwestern regions with low soil salinity, the SPAD values of winter wheat were higher. This result reflects the regional characteristics of the coastal area of the YRD, which is consistent with previous research results [64,65].

## 5. Conclusions

In this paper, the core region of the YRD is taken as the study area, and the winter wheat in the reviving stage is taken as the research object. Through the combination of field measured data, UAV multispectral imagery and S2A multispectral imagery to construct the SPAD inversion model of the winter wheat, a scale-up inversion application was carried out. A multi-scale inversion of the SPAD values of winter wheat in the reviving stage was realized. The main conclusions are as follows:
(1)The SPAD characteristic bands of winter wheat in the reviving stage are the Bg, Br, Breg and Bnir bands. The sensitive vegetation indexes are TCARI, etc., and the characteristic spectral parameters are DVI + TCARI, DVI + OSAVI + TCARI, etc.(2)Based on UAV imagery, the best inversion model was Y=46.803+79.564×G×R−35.583×G×R×REG+33.52×(DVI+TCARI)−9.308×(OSAVI+TCARI), and Y represents the SPAD inversion value. The model is constructed via the multiple linear regression method, with a modeling precision of R^2^ = 0.926, RMSE = 0.63 and MAE = 0.92 and a validation precision of R^2^ = 0.934, RMSE = 0.78 and MAE = 0.87. The model has a strong predictive ability and high universality and is suitable for the SPAD inversion of winter wheat in the reviving stage in the study area.(3)The best SPAD inversion model based on UAV imagery cannot be directly applied to SA imagery for scale-up inversion, so the reflectivity correction of S2A imagery is needed first. The correction coefficients of reflectivity of Bg, Br, Breg and Bnir bands were 0.640638, 0.672978, 0.712553 and 0.796514, respectively.(4)SPAD values of winter wheat in the reviving stage in the study area were concentrated in the range of 40~60, and they showed an increasing trend from the east coast to the southwest and the west, which was strongly correlated with the degree of regional soil salinization.


In this paper, an effective method of integrated satellite-UAV-ground inversion in the study area is proposed, a high-precision and universal SPAD inversion model of winter wheat in the reviving stage is constructed, and a method of UAV-based model applying to S2A images is proposed that realizes real-time, rapid and accurate monitoring of the winter wheat SPAD values at different scales in the study area. These results are of great importance to the monitoring and management of winter wheat in the study area.

## Figures and Tables

**Figure 1 sensors-19-01485-f001:**
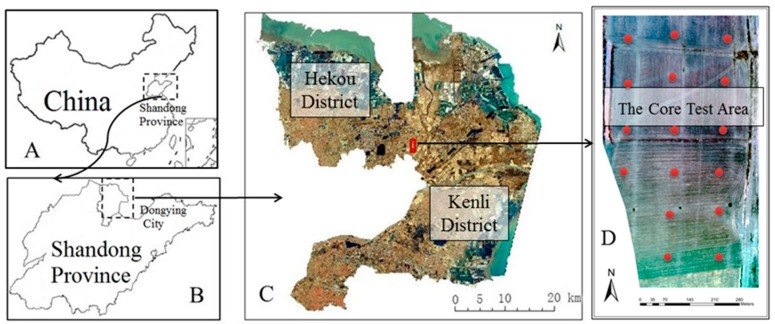
Location of the study area. (A: China; B: Shandong Province; C: the study area; D: part of the core test area, the red dots represent the sampling points).

**Figure 2 sensors-19-01485-f002:**
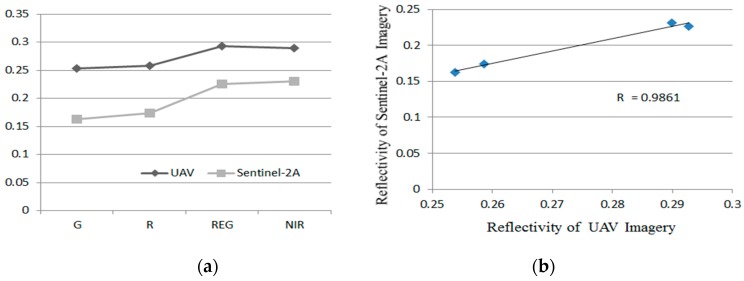
Comparison of surface reflectance between UAV and S2A imagery.

**Figure 3 sensors-19-01485-f003:**
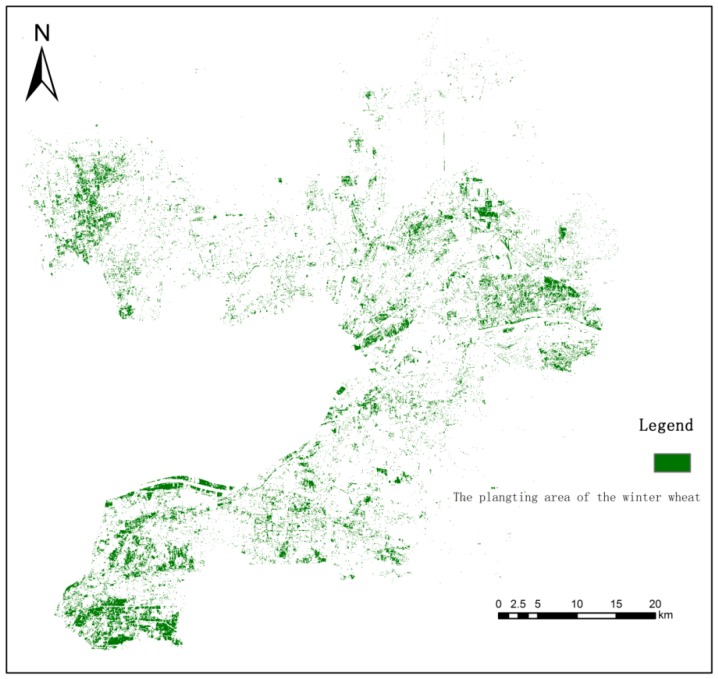
Distribution of winter wheat in the study area in 2018.

**Figure 4 sensors-19-01485-f004:**
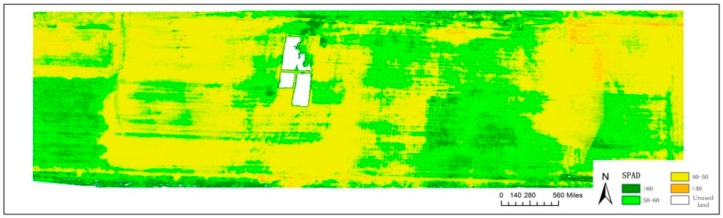
The spatial distribution map of SPAD values in part of the core test area.

**Figure 5 sensors-19-01485-f005:**
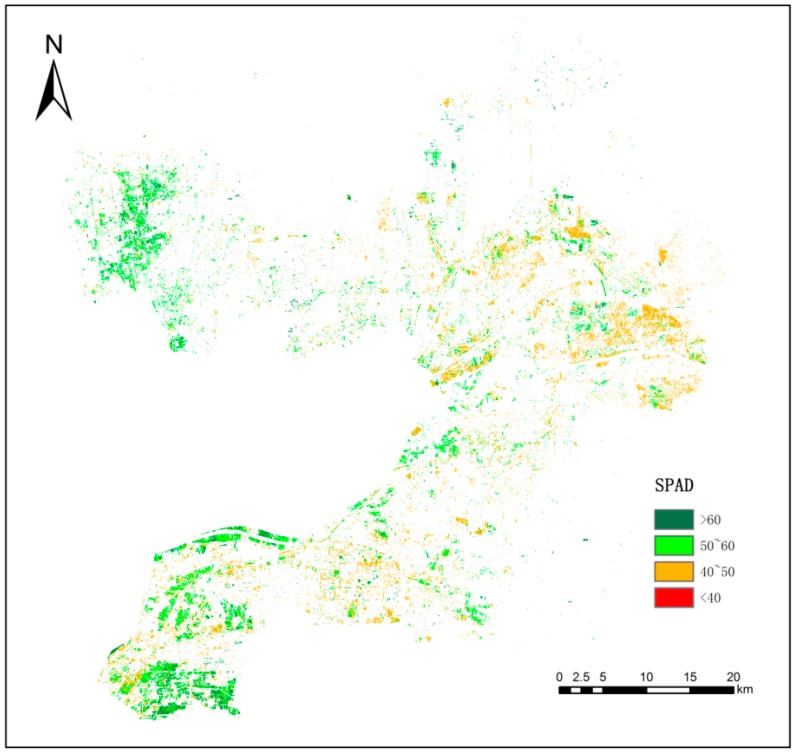
The spatial distribution map of the inverted SPAD values in the study area.

**Table 1 sensors-19-01485-t001:** The corresponding relationship between S2A data and UAV data.

Name of Bands	Bands	S2A		Sequoia (UAV)	
		Central Wavelength (nm)	Spatial Resolution (m)	Bands	Central Wavelength (nm)
Bg	B3-Green	560	10	Green	550
Br	B4-Red	665	10	Red	660
$Bre_{-705}$	B5-Vegetation Red Edge	705	20	--	--
Bre	B6-Vegetation Red Edge	740	20	Red Edge	735
Bnir	B7-Vegetation Red Edge	783	20	Near IR	790

**Table 2 sensors-19-01485-t002:** Vegetation indexes.

Vegetation Indexes	Full Name	Calculation Formulas
**NDVI**	Normalized difference vegetation index	(Bnir−Br)/(Bnir+Br)
**DVI**	Difference vegetation index	Bnir−Br
**RVI**	Ratio vegetation index	Bnir/Br
**SAVI**	Soil adjusted vegetation index	1.5×(Bnir−Br)/(Bnir+Br+0.5)
**OSAVI**	Optimized soil adjusted vegetation index	1.16×(Bnir−Br)/Bnir+Br+0.16)
**CI_red edge_**	Red edge chlorophyll index	(Bre/Br)−1
**TVI**	Triangular vegetation index	0.5×[120×(Bre−Bg)−200(Br−Bg)]
**MCARI**	Modified chlorophyll absorption ratio index	[(Bre−Br)−0.2(Bre−Bg)]×(Bre/Br)
**TCARI**	Transformed chlorophyll absorption ratio index	$3\left[\left(Bre-Br\right)-0.2\left(Bre-Bg\right)\times \left(\frac{Bre}{Br}\right)\right]$

**Table 3 sensors-19-01485-t003:** Correlation between spectral reflectance of UAV and S2A imagery with SPAD values.

Name of Bands	UAV		S2A	
	Bands	R	Bands	R
Bg	Green	0.781 **	B3-Green	0.583 **
Br	Red	−0.803 **	B4-Red	−0.505 **
$Bre_{-705}$	--	--	B5-Vegetation Red Edge	0.628 **
Bre	Red Edge	0.797 **	B6-Vegetation Red Edge	0.587 **
Bnir	Near IR	0.727 **	B7-Vegetation Red Edge	0.529 **

Note: ** significant at the 0.01 probability level.

**Table 4 sensors-19-01485-t004:** Correlation between UAV-based vegetation indexes and SPAD values.

Vegetation Indexes	NDVI	DVI	SAVI_L=0.5_	OSAVI	TCARI	RVI	TVI	MCARI	CI_red edge_
**R**	0.901 **	0.909 **	0.901 **	0.903 **	0.912 **	0.878 **	0.811 **	0.859 **	0.873 **

Note: ** significant at the 0.01 probability level.

**Table 5 sensors-19-01485-t005:** Correlation between UAV-based characteristic spectral parameters and SPAD values.

Spectral Parameters	DVI + TCARI	DVI + OSAVI + TCARI	OSAVI + TCARI	G×R	G×R×REG
**R**	0.916 **	0.915 **	0.914 **	0.86 **	0.841 **

Note: ** significant at the 0.01 probability level.

**Table 6 sensors-19-01485-t006:** Non-linear regression inversion models of SPAD based on UAV imagery.

Method	Parameters		Modeling Precision			Verification Precision		
			R^2^	RMSE	MAE	R^2^	RMES	MAE
**SVR**	Knernel RBF	Gamma = 2	0.929	1.303	0.964	0.864	1.338	1.365
	C = 1.0	Epslion = 0.001						
**BP-NN**	L = 0.3	E = 20	0.919	1.202	1.254	0.917	1.503	1.334
	N = 500	H = 3						
**PLS**	--		0.868	2.044	6.01	0.819	2.707	7.293

**Table 7 sensors-19-01485-t007:** Linear regression inversion models of SPAD based on UAV imagery.

Independent Variables	Method	Formulas	Modeling Precision			Verification Precision		
			R^2^	RMSE	MAE	R^2^	RMES	MAE
**Characteristic Bands**	OLR	Y=−1.3794+83.9773×Bg	0.32	11.23	5.64	0.35	11.24	5.41
	MLR	Y=33.523+50.565×Bg−72.251×Bg+38.49×Bre+49247×Bnir	0.864	2.56	2.32	0.935	0.81	3.18
**Sensitive Vegetation Indexes**	OLR	Y=49.7005+112.2358×DVI	0.35	11.11	5.21	0.34	11.36	5.22
	MLR	Y=52.449−116.267×SAVI+322.353×DVI+24.509×TCARI	0.862	2.03	1.95	0.928	1.29	1.31
**Characteristic Spectral Parameters**	OLR	Y=50.1087+27.582×(DV+TCARI)	0.35	11.1	5.24	0.33	10.49	5.09
	MLR	Y=46.803+79.564×Bg×Br−35.583×Bg×Br×Bre+33.520×(DVI+TCARI)−9.308×(OSAVI+TCARI)	0.926	0.63	0.92	0.934	0.78	0.87

**Table 8 sensors-19-01485-t008:** Reflectance correction coefficient of S2A.

Name of Bands	Bg	Br	Bre	Bnir
Reflectance correction coefficients	0.640638	0.672978	0.772553	0.796514

**Table 9 sensors-19-01485-t009:** Validation of SPAD inversion model.

Cases	Model	S2A Imagery	Formulas	Modeling Precision			Validation Precision		
				R^2^	RMSE	MAE	R^2^	RMSE	MAE
1	Based on S2A imagery	without reflectance correction	$Y=256.22-2648.173\times R+2923\times G+1798.248\times {\mathrm{REG}}_{-705}-1472.229\times \mathrm{REG}-25.852\times \mathrm{NIR}-663.251\times \mathrm{DVI}-308.515\times \mathrm{RVI}+974.188\times \mathrm{GNDVI}+152.059\times \mathrm{MCARI}+297.535\times \mathrm{CI}$	0.849	10.21	7.77	0.845	11.56	6.38
2	Based on UAV imagery	without reflectance correction	Y=46.803+79.564×G×R−35.583×G×R×REG+33.52×(DVI+TCARI)−9.308×(OSAVI+TCARI)	0.926	0.63	0.92	0.605	1.74	1.89
3	Based on UAV imagery	after reflectance correction	the same as case 2	0.926	0.63	0.92	0.905	0.97	1.02

**Table 10 sensors-19-01485-t010:** Precision of winter wheat area extraction.

	Statistical Area (hm^2^)		2015~2016 Average Area (hm^2^)	Extracted Area of 2018 (hm^2^)	Precision (%)
	2015	2016			
Hekou District	5980	5567	5773.5	6033.8	95.7
Kenli District	7605	9772	8688.5	8935.4	97.2
Total	13,585	15,339	14,462	14,969.2	96.6

Note: The statistical area of 2015 and 2016 in the table comes from <Dongying Statistical Yearbook 2017>.

**Table 11 sensors-19-01485-t011:** The proportion of areas with SPAD values in different ranges (without non-agricultural land).

Ranges of SPAD	<40	40~50	50~60	>60
Proportion (%)	0.959	61.913	30.823	6.305

**Table 12 sensors-19-01485-t012:** The results of the inverted SPAD values in the study area.

SPAD	<40	40–50	50–60	>60
Hekou District	0.04%	43.45%	47.32%	9.18%
Kenli District	0.06%	54.52%	37.25%	8.17%
The study area	0.05%	48.98%	42.29%	8.68%

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
