# Peer review of "Integrated Satellite, Unmanned Aerial Vehicle (UAV) and Ground Inversion of the SPAD of Winter Wheat in the Reviving Stage"

_sensors, 2019, doi:10.3390/s19071485_

Reviewer 1 Report

The manuscript presents a study on the combination of satellite data, UAV images and ground data to estimate chlorophyll content in wheat. The article presents some interesting results, but there are many flaws that need to be addressed.

Some major issues:

- Language needs some serious work. There are many grammatical errors, punctuation is frequently incorrect, and most sentences are oddly constructed, with numerous nearly unintelligible excerpts.

- UAVs and satellites have been thoroughly explored as means for quantifying chlorophyll and related variables, like Nitrogen. As a result, new publications on the subject are only justifiable if they bring something new, either by improving upon its predecessors or by bringing some new valuable insights on the problem. Such novelty should be clearly highlighted in the introduction, which is not the case here. The authors mention that they do combine satellite, UAV and ground data, but do not offer any thoughts on how this strategy compares with its predecessors and how it contributes to improve chlorophyll quantification. A better introductory contextualization is needed to guide the reader’s expectations.

- Section 2 as a whole is verbose, confusing, and not detailed enough for experimental reproduction. In addition, more details about the UAV imagery should be provided: what were the weather conditions when the images were capture? What was the angle of solar incidence? How many images were captured? Did the authors experience mosaicking errors? How many missions were carried out? Were they carried out in different days? Did weather conditions vary from day to day?

- Most text in the Results section is unnecessary, as the authors simply repeat what is already clearly shown in the figures and tables. The text should simply synthesize the information and highlight major trends, avoiding too much redundancy.

- The Discussion section is fine in terms of content, but as for the rest of the text, language needs improvement.

Some minor issues that also need addressing:

- Lines 29-30: this excerpt does not make sense and should be rewritten.

- Lines 41-44: very confusing excerpt.

- Lines 50-64: this paragraph is very hard to read. The structure “Author A thinks this…, author B said that, author C showed that…, etc.” is not appropriate. A more suitable way to present relevant information would something like: “The green and red bands have been shown to be valuable for the detection of chlorophyll (author A, author B, etc.). However, some studies have also pointed out the blue channel as a good candidate (author C). Etc, etc.”

- What are the red dots in Figure 1?

- Lines 124-126: the data correspondence has to be better explained. Because UAV images have a much higher spatial resolution than the satellite data, the area contained in one pixel of the satellite images will be represented by a large number of pixels in the UAV images. Thus, saying that ground data represents a unique pixel in both UAV and satellite images does not make much sense.

- Line 142: the term “continuously” implies that images were captured 24 hours a day, which is obviously not the case. Please remove the word.

- Lines 162-163: again, the correspondence between ground data and UAV images has to be better explained. Did each SPAD measurement serve as reference for multiple pixels in the UAV images? If so, how were those selected?

- Lines 164-172: this entire paragraph could be synthesized into a single sentence.

Author Response

Thank you for handling our manuscript entitled “Integrated Satellite, Unmanned Aerial Vehicle (UAV) and Ground Inversion of the SPAD of winter wheat in reviving stage” (Manuscript ID: sensors-463640). We appreciate the comments from the reviewers, which helped to improve the manuscript significantly. In the following section, we explain in detail how we responded to each of the comments.

Response to Reviewer 1 Comments

Point 1: The manuscript presents a study on the combination of satellite data, UAV images and ground data to estimate chlorophyll content in wheat. The article presents some interesting results, but there are many flaws that need to be addressed.

Response 1: We are very grateful for your recognition of this research. We will seriously modify the deficiencies of the article. We are very grateful to the editors and reviewers for your hard work.

Point 2: Language needs some serious work. There are many grammatical errors, punctuation is frequently incorrect, and most sentences are oddly constructed, with numerous nearly unintelligible excerpts.

Response 2: We agreed with the reviewer’s comment about language. Language editing has been done by American Journal Experts as well as a native english-speaking expert.

Point 3: UAVs and satellites have been thoroughly explored as means for quantifying chlorophyll and related variables, like Nitrogen. As a result, new publications on the subject are only justifiable if they bring something new, either by improving upon its predecessors or by bringing some new valuable insights on the problem. Such novelty should be clearly highlighted in the introduction, which is not the case here. The authors mention that they do combine satellite, UAV and ground data, but do not offer any thoughts on how this strategy compares with its predecessors and how it contributes to improve chlorophyll quantification. A better introductory contextualization is needed to guide the reader’s expectations.

Response 3: Thanks for the reviewer's reminding, and we add our thoughts on how this strategy compares with its predecessors and how it contributes to improve chlorophyll quantification in Page 3 line 89-93 and Page 3 line 98-103.

Page 3 line 89-93: Scholars have also conducted research on the application of multisource data and generated robust results [26-28]. However, most of them combined a variety of satellite data with ground measured data [29-31] or satellite images combined with surface hyperspectral data [32,33]. Accordingly, the integration of satellite, UAV and ground data is rare, feasible and is thus worthy of further research.

Page 3 line 98-103: Compared with the traditional measurement method of SPAD values, this study provides a multi-scale, high-efficiency and nondestructive estimation method of SPAD values [34,35], and this method has a higher accuracy and is less affected by the weather compared with being based on the measured data and satellite images [36-38]; moreover, when compared with using measured data and UAV images, the estimation scale of this method is larger [39,40].

Point 4: Section 2 as a whole is verbose, confusing, and not detailed enough for experimental reproduction. In addition, more details about the UAV imagery should be provided: what were the weather conditions when the images were capture? What was the angle of solar incidence? How many images were captured? Did the authors experience mosaicking errors? How many missions were carried out? Were they carried out in different days? Did weather conditions vary from day to day?

Response 4: Thanks for the reviewer's reminding, we provide details about the UAV imagery in Page 4 line 152-159, then, supplement and modify the entire Section 2.

Page 4 line 152-159: On March 8, 2018 at 12:00 to 14:00, the multispectral imagery was acquired by the UAV in the core test area. The weather was sunny, with a temperature range from 4~6 and northwest winds of 3~5 m/s, and the angle of the solar incidence was 52 °. There were three missions; the flight height was 100 m, the flight speed was 5 m/s, the camera shooting time interval was set to 1.5 s, the area was approximately 1 km2, and the flight plan was designed to ensure 80 % overlap with the resolution of 4~5 cm. More than 5,000 valid images were taken. In the later stage, Pix4D software was used to preprocess the UAV imagery, with such activities as mosaic and radiation corrections.

Point 5: Most text in the Results section is unnecessary, as the authors simply repeat what is already clearly shown in the figures and tables. The text should simply synthesize the information and highlight major trends, avoiding too much redundancy.

Response 5: Thanks for the reviewer's reminding. The unnecessary text in the Section 3 has been deleted, and we synthesize the information according to the reviewer's requirements.

Point 6: The Discussion section is fine in terms of content, but as for the rest of the text, language needs improvement.

Response 6: We are very grateful for your recognition of the Discussion section. We have seriously improved the language of the rest of the text.

Point 7: Lines 29-30: this excerpt does not make sense and should be rewritten.

Response 7: Page 1 Lines 27-29: As suggested by the reviewer, we rewrote this part, and the excerpt does not make sense has been deleted.

Point 8: Lines 41-44: very confusing excerpt.

Response 8: As suggested by the reviewer, we rewrote this part to make it easier to understand.

Page 1 Lines 40-43:  Chlorophyll is an important pigment in plant photosynthesis, as its content reveals the ability of a plant to conduct material energy exchange with the external environment, as well as growth status, primary productivity, carbon sequestration ability and nitrogen utilization efficiency, and so on.

Point 9: Lines 50-64: this paragraph is very hard to read. The structure “Author A thinks this…, author B said that, author C showed that…, etc.” is not appropriate. A more suitable way to present relevant information would something like: “The green and red bands have been shown to be valuable for the detection of chlorophyll (author A, author B, etc.). However, some studies have also pointed out the blue channel as a good candidate (author C). Etc, etc.”

Response 9: As suggested by the reviewer, we changed the way we expressed it.

Page 2 Lines 49-52: The green and red bands have been shown as the most valuable for detecting chlorophyll [3,4]; however, some studies have also identified the near-infrared channel as an effective candidate [3,5]. Previous studies also show that reflectance spectra in the visible region (400~700 nm) can effectively estimate chlorophyll [6].

Page 2 Lines 55-59: The red-edge parameters became one of the most important index for the growth of crops [7], as well as for estimating chlorophyll content in plants [8]. And then, the optimal red-edge parameters were screened out by detecting the hyperspectral values and chlorophyll content, and a relationship model was established for them [9].

Page 2 Lines 71-74: The red-edge bands of Sentinel series satellites were used to monitor the chlorophyll content of crops [16]; Sentinel-2 imagery were used to estimate the canopy chlorophyll content and obtained a high accuracy [17].

Point 10: What are the red dots in Figure 1?

Response 10: The red dot in Figure 1 represents the sampling points, and we have added an explanation to this in Page 4 Lines 136-136.

Point 11: Lines 124-126: the data correspondence has to be better explained. Because UAV images have a much higher spatial resolution than the satellite data, the area contained in one pixel of the satellite images will be represented by a large number of pixels in the UAV images. Thus, saying that ground data represents a unique pixel in both UAV and satellite images does not make much sense.

Response 11: As suggested by the reviewer, we explained the data correspondence in more detail. As described in Page 3 Lines 112-117 and Page 6 Lines 211-212.

Page 3 Lines 112-117: the data sources in this paper include satellite, UAV and ground measured data. When building the UAV-based model, a ground sampling point data corresponds to a pixel of UAV imagery. When building the satellite-based model, a ground sampling point data corresponds to a pixel of satellite imagery. When calculating the correction coefficient of reflectivity of the satellite images, there are approximately 1600 UAV image pixels corresponding to a satellite image pixel in the same area.

Page 6 Lines 212-213: The average reflectivity of each band of the 1600 pixels of UAV images and the corresponding pixel of S2A images is obtained, and the relationship between them is analyzed.

Point 12: Line 142: the term “continuously” implies that images were captured 24 hours a day, which is obviously not the case. Please remove the word.

Response 12: Thanks for the reviewer's reminding, and we remove the word “continuously”.

Point 13: Lines 162-163: again, the correspondence between ground data and UAV images has to be better explained. Did each SPAD measurement serve as reference for multiple pixels in the UAV images? If so, how were those selected?

Response 13: Same as Response 11, as suggested by the reviewer, we explained the data correspondence in more detail. As described in Lines 112-117 and Page 6 Lines 212-213.

Page 3 Lines 112-117: the data sources in this paper include satellite, UAV and ground measured data. When building the UAV-based model, a ground sampling point data corresponds to a pixel of UAV imagery. When building the satellite-based model, a ground sampling point data corresponds to a pixel of satellite imagery. When calculating the correction coefficient of reflectivity of the satellite images, there are approximately 1600 UAV image pixels corresponding to a satellite image pixel in the same area.

Page 6 Lines 212-213: The average reflectivity of each band of the 1600 pixels of UAV images and the corresponding pixel of S2A images is obtained, and the relationship between them is analyzed.

Point 14: Lines 164-172: this entire paragraph could be synthesized into a single sentence.

Response 14: Page 5 Lines 175-178: As suggested by the reviewer, this entire paragraph has been synthesized into a single sentence.

Thanks again.

Reviewer 2 Report

- Check spacing between reference and words, such as with "[9]screened". "Jiang et al[7]." would have no period after it as it is part of a sentence. Many of the references have some spacing or type before or after it.

- Have a sentence between headings to introduce sub-sections to follow. For example, after section 2, have a sentence to describe the sub-sections to follow, before having sub-section 2.1. Similar additions to the other sub-sections are suggested too.

- Don't use first person such as we, our, us, etc.

- Check that there is a space between all values and units. e.g. "(0.1km×0.5km)" to be "(0.1 km × 0.5 km)"

- Section 4 heading should be "Discussion" instead of "Discuss"

In general, check spacings between text and special characters thoughout the paper

Author Response

Thank you for handling our manuscript entitled “Integrated Satellite, Unmanned Aerial Vehicle (UAV) and Ground Inversion of the SPAD of winter wheat in reviving stage” (Manuscript ID: sensors-463640). We appreciate the comments from the reviewers, which helped to improve the manuscript significantly. In the following section, we explain in detail how we responded to each of the comments.

Response to Reviewer 2 Comments

Point 1: Check spacing between reference and words, such as with "[9]screened". "Jiang et al[7]." would have no period after it as it is part of a sentence. Many of the references have some spacing or type before or after it.

Response 1: I'm sorry for these simple mistakes, we have checked all the mistakes and corrected them.

Point 2: Have a sentence between headings to introduce sub-sections to follow. For example, after section 2, have a sentence to describe the sub-sections to follow, before having sub-section 2.1. Similar additions to the other sub-sections are suggested too.

Response 2: As suggested by the reviewer, we added sentences between headings to introduce sub-sections. Page 3 Lines 112-117; Page 7 Lines 250-251.

Point 3: Don't use first person such as we, our, us, etc.

Response 3: Thanks for the reviewer's reminding, and we corrected the mistakes.

Point 4:Check that there is a space between all values and units. e.g. "(0.1km×0.5km)" to be "(0.1 km × 0.5 km)".

Response 4:Thanks for the reviewer's reminding, we checked and corrected the mistakes.

Point 5: Section 4 heading should be "Discussion" instead of "Discuss".

Response 5: Thanks for the reviewer's reminding. As suggested by the reviewer, “Discuss” was replaced by “Discussion”.

Point 6: In general, check spacings between text and special characters thoughout the paper.

Response 6: Thanks for the reviewer's reminding. As suggested by the reviewer, we checked and corrected the spacings between text and special characters thoughout the paper.

Thanks again.

Reviewer 3 Report

Dear Authors, 

you have presented Integrated Satellite, UAV and Ground Inversion of the SPAD in order to express the relation content of chlorophyll.of winter wheat. Here are some comments:

1) The state-of-the art comparison with other methods used for to determine the chlorophyll.of winter wheat and the methods based on UAV and satellite imagery used in agriculture is missing.

2) Coefficient of determination (R2), root mean square error (RMSE) and mean absolute error  (MAE) are usually used for the evaluate the performance. Please add.

3) Abstract is overly verbose and the contribution of the work as well as its innovative aspect is not well understood.

4) You say that the best inversion model you get with a multiple regression.  But you should provide more evidence on this also experimentally. 

5) Please add a statistical analysis on the correlation made.

Author Response

Thank you for handling our manuscript entitled “Integrated Satellite, Unmanned Aerial Vehicle (UAV) and Ground Inversion of the SPAD of winter wheat in reviving stage” (Manuscript ID: sensors-463640). We appreciate the comments from the reviewers, which helped to improve the manuscript significantly. In the following section, we explain in detail how we responded to each of the comments.

Response to Reviewer 3 Comments

Point 1: The state-of-the art comparison with other methods used for to determine the chlorophyll of winter wheat and the methods based on UAV and satellite imagery used in agriculture is missing.

Response 1: Thanks for the reviewer's reminding, and we add our thoughts on how this strategy compares with its predecessors and how it contributes to improve chlorophyll quantification in Page 3 line 89-93 and Page 3 line 98-103.

Page 3 line 89-93: Scholars have also conducted research on the application of multisource data and generated robust results [26-28]. However, most of them combined a variety of satellite data with ground measured data [29-31] or satellite images combined with surface hyperspectral data [32,33]. Accordingly, the integration of satellite, UAV and ground data is rare, feasible and is thus worthy of further research.

Page 3 line 98-103: Compared with the traditional measurement method of SPAD values, this study provides a multi-scale, high-efficiency and nondestructive estimation method of SPAD values [34,35], and this method has a higher accuracy and is less affected by the weather compared with being based on the measured data and satellite images [36-38]; moreover, when compared with using measured data and UAV images, the estimation scale of this method is larger [39,40].

Point 2: Coefficient of determination (R2), root mean square error (RMSE) and mean absolute error  (MAE) are usually used for the evaluate the performance. Please add.

Response 2:As suggested by the reviewer, we added Coefficient of determination (R2) and root mean square error (RMSE) in Page 6 Lines 197-199, Lines 202-206, Page 9 Table 6, Page 9 Lines 290-301 and Page 10-11 Table 8.

Page 6 Lines 197-199: The modeling precision of the model was synthesized with the verification precision (represented by decision coefficient R2, root mean squared error RMSE and mean absolute error MAE) to select the best inverse model.

Page 6 Lines 203-209: RMES represents the degree of dispersion between the data and the true values; the lower the RMES is, the lower the degree of dispersion of the data is, and the higher the accuracy is [50]. MAE can accurately reflect the actual error by avoiding offset errors, the lower the MAE is, the lower the error of the model is, and the higher the precision is [51,52]. Relevance analysis and model building are performed by software SPSS 22 (Statistical Product and Service Solutions) and WAKE (Waikato Environment for Knowledge Analysis).

Page 9 Table 6

(The table is displayed in the WORD)

Page 9 294-306: From the perspective of modeling precision, the models based on multiple linear regression method are higher than that based on a linear regression method. Based on the multiple linear regression method, the R2 of the models based on the three types of variables are above 0.8, which indicate high prediction ability, however, AME and RMSE show significant differences due to different independent variables. Overall, the model constructed with spectral parameters as independent variables, based on the multiple linear regression method, has the highest modeling R2, the lowest modeling RMSE and MAE, which shows high accuracy and stability. The same trend is reflected of verification precision. Therefore, Y=46.803+79.564×G×R-35.583×G×R×REG+33.52×(DVI+TCARI) -9.308×(OSAVI+TCARI)is chosen as the best SPAD inversion model (Y represents the inversion values of the SPAD of winter wheat in reviving stage), the modeling precisions are R2 = 0.926, RMSE = 0.63 and AME = 0.92, and the verification precisions are R2 = 0.934, RMSE = 0.78 and AME = 0.87.

Page 10-11 Table 8

(The table is displayed in the WORD)

Point 3: Abstract is overly verbose and the contribution of the work as well as its innovative aspect is not well understood.

Response 3: Thanks for the reviewer's reminding, and we have reduced and adjusted the content of the abstract and reduced it.

Point 4: You say that the best inversion model you get with a multiple regression.  But you should provide more evidence on this also experimentally.

Response 4: As suggested by the reviewer, we added the models based on a linear regression and the relevant contents of the Section 2 and Section 3 are modified.

As for the Support Vector Machine (SVM) and other nonlinear regression methods, WEKA software was used to try in this study, and the R2 of the best model was about 0.9, RMSE was about 0.5, and AME was about 0.5. But the modeling method had encapsulation, and the model constructed by multiple linear regression method could reach the same accuracy standard. Therefore, no matter from practicability, intuition and experimental reproduction, multiple linear regression method is the best choice for this research. Since the regression method is not the focus of this paper, this reflection is not reflected in the discussion. Please forgive.

Page 6 Lines 186-188: Finally, the characteristic bands, sensitive vegetation indexes and characteristic spectral parameters of the modeling set were used as dependent variables to construct the SPAD inversion model by various regression methods.

Page 9 Table 6

(The table cannot be displayed)

Page 9 294-295: From the perspective of modeling precision, the models based on multiple linear regression method are higher than that based on a linear regression method.

Point 5: Please add a statistical analysis on the correlation made.

Response 5: As suggested by the reviewer, we strengthen the statistical analysis in this paper.

Thanks again.

Round  2

Reviewer 1 Report

My concerns were properly addressed and, as a result, the manuscript is vastly improved. I do not have any further suggestions or observations.

Author Response

Thank you for handling our manuscript entitled “Integrated Satellite, Unmanned Aerial Vehicle (UAV) and Ground Inversion of the SPAD of winter wheat in reviving stage” (Manuscript ID: sensors-463640). We appreciate the comments from the reviewers, which helped to improve the manuscript significantly. In the following section, we explain in detail how we responded to each of the comments.

Response to Reviewer 1 Comments

Point 1: My concerns were properly addressed and, as a result, the manuscript is vastly improved. I do not have any further suggestions or observations.

Response 1: We are very grateful for your recognition of this research. We are very grateful to the editors and reviewers for your hard work.

Reviewer 3 Report

p.p1 {margin: 0.0px 0.0px 0.0px 0.0px; font: 12.0px 'Helvetica Neue'; min-height: 14.0px} p.p2 {margin: 0.0px 0.0px 0.0px 0.0px; font: 12.0px 'Helvetica Neue'}

The paper was slightly improved but there are still some open points: 

Point 3 - The abstract is still there and must be rewritten and shortened. An abstract summarizes, usually in one paragraph of 300 words or less, the major aspects of the entire paper in a prescribed sequence that includes: 1) the overall purpose of the study and the research problem(s) you investigated; 2) the basic design of the study; 3) major findings or trends found as a result of your analysis; and, 4) a brief summary of your interpretations and conclusions.

Also point 4 need to be better commented. 

Author Response

Thank you for handling our manuscript entitled “Integrated Satellite, Unmanned Aerial Vehicle (UAV) and Ground Inversion of the SPAD of winter wheat in reviving stage” (Manuscript ID: sensors-463640). We appreciate the comments from the reviewers, which helped to improve the manuscript significantly. In the following section, we explain in detail how we responded to each of the comments.

Response to Reviewer 3 Comments

Point 1:

p.p1 {margin: 0.0px 0.0px 0.0px 0.0px; font: 12.0px 'Helvetica Neue'; min-height: 14.0px}

p.p2 {margin: 0.0px 0.0px 0.0px 0.0px; font: 12.0px 'Helvetica Neue'}

Response 1: Thanks for the reviewer's reminding, and we have consulted the editor, and the following is the editor's reply:

Please ignore the first suggestion. Our production team will help format
the paper after it is accepted for publication.
Please just revise the paper according to the suggestions related to the
research content of the article.

We will pay more attention to the format in the future.

Point 2: Point 3 - The abstract is still there and must be rewritten and shortened. An abstract summarizes, usually in one paragraph of 300 words or less, the major aspects of the entire paper in a prescribed sequence that includes: 1) the overall purpose of the study and the research problem(s) you investigated; 2) the basic design of the study; 3) major findings or trends found as a result of your analysis; and, 4) a brief summary of your interpretations and conclusions..

Response 2: As suggested by the reviewer, the abstract has been rewritten and shortened to 290 words. Page 1 Line 12-33.

Point 3: Also point 4 need to be better commented.

Response 3: As suggested by the reviewer, more experimentally evidence has been provided. Page 9 Line 289-293, Table 6, Page 9 Line 295-296, Table 7 and Page 10 Line 298-316.

Page 9 Line 289-293:

The results shown in Table 6 are inversion models of SPAD based on non-linear regression methods, as follows, support vector regression (SVR), back propagation neural network (BP-NN), partial least squares regression (PLS). And the results shown in Table 7 are inversion models of SPAD based on one-variable linear regression (OLR) and multiple linear regression (MLR) methods.

Table 6:

Table 6. Non-linear regression inversion models of SPAD based on UAV imagery

(the table will be displayed in WORD)

Page 9 Line 295-296:

Objectively speaking, models based on non-linear methods have good prediction ability, lower error and stronger stability. SVR has the highest accuracy.

Table 7:

Table 7. Linear regression inversion models of SPAD based on UAV imagery

(the table will be displayed in WORD)

Page 10 Line 298-316:

As for the linear regression methods, the modeling precisions of MLR, are higher than that of OLR. The modeling R2 of the models using MLR, based on the three types of variables, are above 0.8, which indicate high prediction ability; however, AME and RMSE show significant differences due to different independent variables. Overall, the model constructed with spectral parameters as independent variables, based on MLR, has the highest modeling R2, the lowest modeling RMSE and MAE, which shows high accuracy and stability. The same trend is also reflected of verification precisions. Therefore, the model constructed by using MLR method with spectral parameters as independent variables is the best.

Comparing the inversion models based on SVR and MLR method, although the modeling R2 of the best MLR model is worse than that of SVR model, its RMSE and AME indicate a lower prediction error. And the verification precisions of the best MLR model are better than SVR, which show a stronger stability. Meanwhile, for practical application, the MLR model is simpler, more visible and can provide better experimental repeatability, while encapsulated SVR does not have these advantages. Therefore, no matter from accuracy of predictive, practicability, and experimental repeatability, MLR method is the best choice. The best SPAD inversion model is Y=46.803+79.564×G×R-35.583×G×R×REG+33.52×(DVI+TCARI)-9.308×(OSAVI+TCARI) (Y represents the inversion values of the SPAD of winter wheat in reviving stage) with the modeling precisions of R2 = 0.926, RMSE = 0.63 and AME = 0.92, and the verification precisions of R2 = 0.934, RMSE = 0.78 and AME = 0.87.

Thanks again.
